# Pharmacological Modulations of Nrf2 and Therapeutic Implications in Aneurysmal Subarachnoid Hemorrhage

**DOI:** 10.3390/molecules28041747

**Published:** 2023-02-12

**Authors:** Qia Zhang, Jianmin Zhang, Jun Mo

**Affiliations:** 1Department of Neurosurgery, The Fourth Affiliated Hospital, School of Medicine, Zhejiang University, Yiwu 322000, China; 2Department of Neurosurgery, The Second Affiliated Hospital, School of Medicine, Zhejiang University, Hangzhou 310009, China

**Keywords:** subarachnoid hemorrhage, oxidative stress, antioxidant therapy, Keap1-Nrf2 pathway, brain injury

## Abstract

An aneurysmal subarachnoid hemorrhage (aSAH) is a subtype of stroke with high morbidity and mortality. The main causes of a poor prognosis include early brain injury (EBI) and delayed vasospasm, both of which play a significant role in the pathophysiological process. As an important mechanism of EBI and delayed vasospasm, oxidative stress plays an important role in the pathogenesis of aSAH by producing reactive oxygen species (ROS) through the mitochondria, hemoglobin, or enzymatic pathways in the early stages of aSAH. As a result, antioxidant therapy, which primarily targets the Nrf2-related pathway, can be employed as a potential strategy for treating aSAH. In the early stages of aSAH development, increasing the expression of antioxidant enzymes and detoxifying enzymes can relieve oxidative stress, reduce brain damage, and improve prognosis. Herein, the regulatory mechanisms of Nrf2 and related pharmacological compounds are reviewed, and Nrf2-targeted drugs are proposed as potential treatments for aSAH.

## 1. Introduction

Aneurysmal subarachnoid hemorrhage (aSAH) is a severe subtype accounting for 9.7% of strokes [1]. Despite being less common than ischemic stroke (IS) and intracerebral hemorrhage (ICH), it mainly affects younger patients and leads to the loss of many years of productive life [2]. When aSAH occurs, the direct toxic effect of heme on neuronal cells can lead to oxidative stress, an inflammatory response, and neurological damage in patients, usually with a poor prognosis. Survivors with cognitive impairments frequently experience mood disorders, fatigue, long-term bed rest, comas, and even death [3]. There are still many challenges related to the prevention of primary injury and secondary injury induced by aSAH which must be overcome to achieve the best patient outcome.

Early brain injury (EBI) and delayed cerebral vasospasm (CVS) are the main pathophysiological mechanisms of brain injury in aSAH. Additional investigations have been conducted in light of its etiology, and it was found that oxidative stress plays a pivotal role in the development of both types of pathophysiological process [4,5,6]. In the acute phase of aSAH, blood components primarily enter the subarachnoid space, releasing oxyhemoglobin (oxyhb), which leads to mitochondrial dysfunction and overexpression of peroxidase. This causes excessive reactive oxygen species (ROS) production that exceeds the body’s antioxidant capacity, leading to EBI, including blood–brain barrier (BBB) disruption, neuroinflammation, and neuronal apoptosis, leading to long-term neurological dysfunction [7]. In addition, oxidative stress can also result in degradation of the vascular endothelium, leading to CVS and increasing the risk of delayed cerebral ischemia (DCI). Recent studies have shown that delayed apoptosis, cortical diffusion ischemia, microthrombosis, and microcirculatory disorders are also important causes of DCI [8]. These components are linked together by neuroinflammation and oxidative stress damage following aSAH.

As mentioned above, antioxidant therapy has important value in the treatment of aSAH, including reducing ROS production and eliminating excessive ROS. As the key to the antioxidant function of the body, the expression of antioxidant enzymes is related to oxide content and regulated by different pathways. The nuclear factor-erythroid-2-related factor 2 (Nrf2) and antioxidant response element (ARE) pathways plays pivotal roles in the basal activity and corresponding induction of genes encoding several antioxidant and phase II detoxifying enzymes and related proteins [9]. Hyperactivation of Nrf2 has been shown to promote neurons survival and BBB protection in aSAH and improve long-term prognosis [10]. Therefore, Nrf2-targeted therapy has considerable potential in the treatment of aSAH. In this review, the research progress made in Nrf2 pathway-related drugs is reviewed and summarized, and new insights are provided for the potential treatment of antioxidant drugs.

## 2. The Mechanisms of Nrf2 Modulation

Nrf2 plays a vital role in maintaining cellular homeostasis, especially when cells are exposed to oxidative stress [10]. Nrf2 target genes include antioxidant enzymes and detoxification enzymes such as heme oxygenase-1 (HO-1), glutathione S-transferase (GST), NADP(H), and quinine oxidoreductase (NQO) [11]. In animal models of aSAH, the Nrf2 is activated and enhanced in the brain, and Nrf2 activation can reduce oxidative stress, neuroinflammation, and BBB disruption [7]. Several Nrf2 inducers have been reported, and the mechanisms of Nrf2 activation include the classical Keap1-dependent pathway as well as the Keap1-independent pathway [9]. Clarification of the regulation of Nrf2 will help further our understanding of cellular defense mechanisms against oxidative stress and may highlight potential Nrf2-targeted therapies for aSAH.

### 2.1. Keap1-Dependent Regulation

Keap1 is the main mechanism responsible for the regulation of Nrf2 [12]. This protein has 624 amino acids with three functional domains, including the broad complex/tramtrack/bric-a-brac (BTB) domain, the intervening region (IVR), and the double glycine/Kelch domain [9]. There are also more than 20 free sulfhydryl groups in the constituent cysteine residues of the BTB and IVR domains. These highly reactive functional groups act as stress sensors, and various oxidative stresses can modify these residues [13,14]. The BTB domain is responsible for the dimerization of the two Keap1 molecules, while the Kelch repeat contains regions that are responsible for binding Nrf2. BTB and Kelch domains are connected via the IVR domain and regulate the activity of Keap1 [9]. Nrf2 contains 605 amino acids divided into seven domains (Neh1-7). Among them, the Neh2 domain at the N-terminal is the main regulatory domain [15]. The Neh2 domain includes seven lysine residues and two binding sites (ETGE and DLG motifs), which are the main binding sites with Keap1, and it participates in the ubiquitination regulation of Nrf2 [16].

Under normal conditions, Keap1 binds to the Neh2 domain of Nrf2 in the cytoplasm and promotes ubiquitin-dependent degradation, thereby maintaining Nrf2 at low levels and preventing constitutive activation of oxidative stress [9]. Each Kelch domain of the Keap1 homodimer is linked to the Nrf2 protein by a low-affinity DLG motif and a high-affinity ETGE motif, with the former having 1/100 of the affinity of the latter [15]. The hinge and latch hypothesis proposes that the ETGE binding site acts as a hinge, while the DLG binding site acts as a latch [16]. When bound at both sites, Nrf2 is perfectly positioned to undergo ubiquitination and subsequent proteasome degradation via the 26S proteasome [9]. When in a state of oxidative stress, the cysteine residue of Keap1 is modified, and this modification causes conformational changes in the protein, resulting in the release of Nrf2 from the low-affinity site and disturbing the transfer of ubiquitin [9]. Thus, the degradation of Nrf2 is reduced and free Nrf2 accumulates in the cytoplasm. Subsequently, Nrf2 translocates into the nucleus and binds to ARE, activating the transcription of antioxidant genes. As an oxidative stress sensor, Keap1 opens up many opportunities to explore the importance of the role of its residues in the regulation of the Nrf2 pathway.

### 2.2. Keap1-Independent Regulation

Although Keap1 plays a dominant role in regulating Nrf2, there is substantial evidence that there are alternative mechanisms used to activate Nrf2 that are independent of Keap1 [17]. The expression and function of Nrf2 can be regulated at multiple levels, including transcriptional, post-transcriptional, protein modification, and subcellular localization. The promoter region of Nrf2 contains DNA binding sites (ARE/XRE), which means that Nrf2 can activate its own expression to form a positive feedback loop, thereby enhancing the cell’s defense against oxidative stress [18]. The miRNAs are also involved in the regulation of Nrf2 protein expression. Studies have shown that mi-144 is negatively correlated with Nrf2, and miR-144 overexpression can decrease the Nrf2 level, reduce glutathione regeneration, and alter the antioxidant capacity of cells [19]. In addition, Nrf2 contains many serine, threonine, and tyrosine residues, and phosphorylation of these residues by various kinases can also regulate the exclusion and degradation of Nrf2. Several kinase pathways that are involved in the regulation of Nrf2 have also been identified. Rojo et al. found that nordihydroguaiaretic acid (NDGA) activates Nrf2 via MAPK and PI3K pathways and identified GSK-3β as an integrator of these pathways and a gatekeeper of Nrf2 stability at the Neh6 phosphorylation level [20]. Huang et al. found that PKC-catalyzed phosphorylation of Nrf2 at Ser-40 is a critical signaling event leading to ARE-mediated cellular antioxidant response [21].

## 3. Nrf2 Activator in the Treatment of aSAH

Nrf2-related aSAH drugs mainly exert their neuroprotective effects by upregulating Nrf2 expression through the Keap1-dependent or Keap1-independent pathway (Figure 1). When activated, Nrf2 is transferred from the cytoplasm to the nucleus. It further initiates gene expression; increases the expression of antioxidant enzymes and detoxification enzymes such as HO-1, GST, NQO, UGT, γ GCS, SOD, etc.; reduces oxidative stress and neuroinflammation; reduces the disruption of the BBB; and improves EBI [22]. At the same time, studies have shown that Nrf2-related pathways can enhance mitophagy and reduce oxidative stress. Upregulation of Nrf2 in endothelial cells can improve vasospasm and reduce the occurrence of DCI. In addition, studies have also shown that Nrf2-related drugs can improve cognitive impairment after aSAH and have certain values for the long-term prognosis of patients with aSAH [23].

### 3.1. Compounds Activate Nrf2 through the Keap1-Dependent Pathway

Under basic conditions, Nrf2 is isolated in the cytosol by the Keap1 homodimer, which promotes ubiquitination of Nrf2 and proteasome degradation. When cells are damaged by chemical or oxidative stress, Keap1 leads to the release of Nrf2 from a Keap1 molecule through conformational changes mediated by its reactive cysteine residues. Nrf2 can no longer be ubiquitinated and degraded, so Keap1 is completely saturated by Nrf2, allowing newly synthesized Nrf2 to accumulate and transport to the nucleus [17]. Pharmacological activation of Nrf2 by various compounds such as allyl sulfides, dithiophenones, flavonoids, isothiocyanates, polyphenols, and triterpenes has been proposed to prevent many diseases related to oxidative stress (Table 1). In the past, there have been many studies on the mechanism of Nrf2 in treatment of various cancers, such as breast cancer [24,25,26,27,28], but the impact of Nrf2 modulation in the treatment of SAH has not been deeply studied. In recent years, researchers have further studied the therapeutic intervention value of Keap1-dependent Nrf2 expression regulatory compounds for aSAH. Moreover, animal studies have shown that implementing drug regulation of Keap1 to increase Nrf2 expression can not only improve the prognosis of aSAH in the acute phase but can also improve the long-term prognosis. Various Keap1-dependent Nrf2 activators have been studied in the treatment of aSAH, including aloperine (ALO), andrographolide (Andro), astaxanthin (ATX), oleanolic acid (Oleanolic), sulforaphane, dimethyl fumarate, and tert-butyl hydroquinone [23,29,30,31,32,33,34]. After the onset of aSAH, these treatments mainly follow the Keap1-Nrf2-ARE pathway and regulate the antioxidant enzymes and detoxification enzymes associated with aSAH, increasing the expression of antioxidant enzymes and detoxification enzymes such as heme oxygenase-1 (HO-1), glutathione S transferase (GST), NADP (H), and quinine oxidoreductase (NQO), and then inhibiting EBIs caused by high levels of ROS production in aSAH.

It should be noted that, in addition to interfering with the expression of related antioxidant enzymes and detoxification enzymes, studies have shown that some Nrf2 inducers acting on the Keap1-dependent pathway can improve cerebral vasospasm after aSAH [35]. When oleanolic acid RTA408 was administered in mice with early aSAH, it alleviated the inhibition of Nrf2 expression in the basilar artery induced by aSAH and inhibited the expression of NF-κ B and iNOS in the basilar artery of mice after aSAH, and then alleviated cerebral vasospasm [35]. Moreover, RTA408 can reduce the inflammatory response in the hippocampus after aSAH. Whether this is related to the long-term cognitive impairment of aSAH patients still requires further investigation. In addition, it has been proven that erythropoietin EPO can also activate the Keap1-Nrf2-ARE pathway. For the animal model of aSAH, erythropoietin EPO promotes the production of endothelial nitric oxide NO from vascular endothelium, promotes the activation of Nrf2, and alleviates the cerebral vasospasm after aSAH [36,37].

In addition to suppressing CVS, the regulation of mitochondrial function has also been one of the focuses of research on prognosis interventions for aSAH in recent years. Mitoquinone (MitoQ) is a mitochondrial-targeted drug [38,39], which can better prevent mitochondrial dysfunction than non-targeted antioxidants. Previous studies have shown that MitoQ can alleviate vascular calcification through the Keap1-Nrf2-ARE pathway [40], which proves that Nrf2 is one of the targets of MitoQ in vivo therapy. Early breakdown of the BBB in aSAH is one of the important mechanisms leading to poor prognosis in aSAH patients [41,42]. Recent studies have shown that mitochondrial dysfunction is the potential cause of BBB breakdown in early brain injury [43], and mitochondrial improvement can prevent neurological deficit after aSAH [44,45]. Therefore, MitoQ has stronger pertinence for EBI treatment after aSAH. Nrf2 has been identified as the main target of BBB destruction in various diseases and aSAH models [39,46], and prohibitin2 (PHB2) is a receptor located in the inner membrane of mitochondria, which is related to mitochondrial dynamics [47,48]. Optic atrophy 1 (OPA1), a protein downstream of PHB2, may improve mitochondrial fusion in the central nervous system, thus playing a neuroprotective role in neurodegeneration [49], and it has been demonstrated that Nrf2 and PHB2 genes are combined and expressed [50]. This experiment proves that MitoQ can weaken the BBB destruction of EBI after aSAH and improve neurological function injury through the Nrf2-PHB2-OPA1 pathway. MitoQ can promote mitophagy through the Keap1-Nrf2-PHB2 pathway, reduce neuronal death related to oxidative stress, and improve brain injury after aSAH [51]. Therefore, promoting mitophagy through the Keap1-Nrf2-PHB2 pathway and then reducing ferroptosis after aSAH may be the main mode of action of Nrf2-related drugs.

EBI following aSAH results in early neurological impairment, and the majority of aSAH patients experience long-term cognitive impairments, which cause the poor prognosis in the late stage. DMF and tBHQ are Nrf2 pathway-related medications that can lessen post-aSAH cognitive impairment. In addition to the increase in antioxidant enzyme expression against ROS overproduction, iron deposition and lipid peroxide accumulation are inhibited, thus inhibiting the ferroptosis process. It is conceivable that medications targeting the Nrf2 pathway may enhance the cognitive performance of aSAH patients via the Keap1-Nrf2-PHB2 pathway, since the improved mitochondrial function can prevent neurological impairment after aSAH. Drugs targeting the Nrf2 pathway not only alleviate EBI following aSAH but also potentially treat progressive cognitive dysfunction in aSAH patients. Additional research is still required to determine the precise therapeutic effect and whether there are any other therapeutic targets besides the enhancement of mitochondrial function.

Therefore, Keap1-dependent-Nrf2-modulating compounds mainly improve the EBI and functional prognosis after aSAH in three different ways: (1) regulation of antioxidant enzyme expression, (2) regulation of mitochondrial function, and (3) inhibiting vasospasm after aSAH.

**Table 1 molecules-28-01747-t001:** Compounds directly acting on Keap1-dependent regulation signaling pathway and their therapeutic effects.

Compounds	Brief Introduction	Therapeutic Effects	Literature
Aloperine (ALO)	Isolated from the legume plant *Sophora alopecuroides* L.	Upregulates the expression of Nrf2 and improves oxidative stress during EBI.	[29]
Andrographolide (Andro)	A diterpenoid of the labdane family extracted from the Asian plant *Andrographis paniculate*.	Increases HO-1 expression and improves oxidative stress during EBI.	[30]
Astaxanthin (ATX)	A carotenoid widely present in algae and aquatic animals.	Increases the expression of antioxidant enzymes and detoxification enzymes such as HO-1, NQO-1, and GST-α 1.	[31,52,53,54,55]
Oleanolic	Oleanolic botanical triterpenoids	Increases HO-1 expression and inhibits ROS production	[32]
Sulforaphane	-	Inhibits neuroinflammation and alleviates cerebral vasospasm after aSAH	[33]
Dimethyl fumarate (DMF)	-	Alleviates oxidative stress and neuroinflammation through the Keap1-Nrf2-ARE pathway	[34]
Tert-butyl hydroquinone (tBHQ)	-	reduces EBI after experimental aSAH by enhancing Nrf2-independent autophagy, in addition to activating Keap1-Nrf2 signaling pathway, and improves cognitive dysfunction.	[23,56]
RTA408	The second-generation semi-synthetic oleanane triterpenes	Alleviates the inhibition of Nrf2 and the expression of NF-κB and iNOS in the basilar artery induced by aSAH; alleviates vasospasm; enhances the antioxidant and anti-inflammatory effects of the Nrf2-ARE pathway, and finally reduces the apoptosis induced by aSAH; improves CVS, and secondary brain injury	[35]
MitoQ	-	Alleviates vascular calcification and enhances mitochondrial autophagy	[40,57]
Melatonin	N-acetyl 5-methoxytryptamine	Induces mitochondrial autophagy and reduces oxidative stress injury in aSAH through the Nrf2-ARE pathway	[58]
Mangiferin (MF)	A natural C-glucoside flavone	Increases HO-1 expression through the Nrf2-related pathway and inhibits neuronal apoptosis and neuroinflammation induced by ROS.	[59]
Erythropoietin (EPO)	-	Increases the expression of HO-1 and promotes the production of NO in the vascular endothelium, reducing CVS.	[60,61]
Salvianolic acid A	Components of Salviae Miltiorrhizae Bunge	Alleviates oxidative stress and neuroinflammation in acute aSAH by regulating the Nrf2-ARE pathway, and alleviates EBI	[39]
Luteolin (LUT)	Flavonoids are widely found in vegetables and fruits	Inhibits the activation of NLRP3 inflammatory corpuscles, which may depend on the upregulation of the Nrf2 signaling pathway	[62]
Astragaloside IV (AS-IV)	A newly found glycoside of cycloartane-type triterpene, is the effective component extracted from *Astragalus membranaceus*	Inhibits ferroptosis in aSAH by activating Nrf2/HO-1 pathway and increasing the levels of GSH, GPX4, and SLC7A11	[63]

(EBI: early brain injury; HO-1: Heme oxygenase-1; NQO-1: quinine oxidoreductase-1; GST-α1: glutathione-S-transferase α1; CVS: cerebral vasospasm; ROS: reactive oxygen species; NLRP3: nucleotide-binding oligomerization domain (NOD)-like receptor family pyrin domain containing 3; NO: nitric oxide; Nrf2: nuclear factor erythroid 2-related factor 2; Keap 1: kelchlike ech-associated protein 1; GSH: Glutathione; GPX4: glutathione peroxidase 4; SLC7A11: solute carrier family 7 member 11).

### 3.2. Compounds Activate Nrf2 through the Keap1-Independent Pathway

#### 3.2.1. AMPK-PGC1α-Nrf2 Pathway

AMP-activated protein kinase (AMPK) is a major regulator of cellular energy metabolism and is involved in epigenetic modifications that control cell differentiation [64], which can be activated by bioactive compounds such as resveratrol and polyphenol-rich foods [65,66,67]. PGC-1α is a downstream molecule of AMPK that is primarily involved in mitochondrial biogenesis [68,69]. Recent studies have found that PGC-1α is associated with various inflammatory and metabolic diseases [70]. Phosphorylation of PGC1 α is an important index of AMPK signaling pathway activation, which is a bridge between the AMPK and Nrf2 signaling pathway [71]. PGC1α can activate transcription factors such as Nrf2 and reduce mitochondrial ROS production, thereby reducing oxidative stress injury [72]. Since mitochondria are the main source of ROS after aSAH, modulating the AMPK-PGC1α-Nrf2 pathway has great potential in the treatment of aSAH (Table 2).

The neuroprotective effects of puerarin have been demonstrated in a variety of central nervous system disorders, including Parkinson’s disease [73], Alzheimer’s disease [74], acute spinal cord injury [75], and cerebral ischemic injury [76,77]. Following aSAH, blood cell degradation products (such as heme) can activate oxidation and lipid peroxidation, thereby triggering ferroptosis [78], which is one of the important pathophysiological mechanisms of EBIs in aSAH [79,80]. Recent studies have shown that puerarin can improve short-term (24 h) and long-term (26 days) neurological deficits after aSAH, primarily through the AMPK-PGC1α-Nrf2 signaling pathway, to alleviate oxidative stress-induced ferroptosis [81].

Interestingly, in the study of the effects of puerarin on oxidative stress injury and photoaging of human fibroblasts induced by UVA radiation, it was found that puerarin can increase the level of antioxidant enzyme mRNA through Keap1-dependent signaling pathway, thus improving the antioxidant capacity of cells, and it can successfully eliminate reactive oxygen species (ROS) and malondialdehyde (MDA) induced by UVA [82]. Therefore, puerarin may regulate antioxidative stress through both Keap1-independent and Keap1-dependent signaling pathway, but whether puerarin plays a role in aSAH treatment through Keap1-dependent signaling pathway remains to be further studied.

**Table 2 molecules-28-01747-t002:** Compounds acting on Keap1-independent regulation signaling pathway and their therapeutic effects.

Pathway	Compounds	Brief Introduction	Therapeutic Effects	Literature
AMPK-PGC1α-Nrf2	Puerarin	A type of flavonoid glycoside, extracted from the pueraria lobata root	Alleviates oxidative stress and iron death after aSAH, and improves neurological functions	[81]
SIRT1-Nrf2	Salvianolic acid B	A natural polyphenolic compound extracted from *Salvia miltiorrhiza*	Inhibits ROS overproduction induced by aSAH, and increases SOD and GSH levels	[83]
	Isoliquiritigenin	The natural flavonoids extracted from licorice	Increases the expression of HO-1, NQO-1, and SOD	[84]

(HO-1: Heme oxygenase-1; NQO-1: Quinine oxidoreductase-1; SOD: Superoxide dismutase; GSH: glutathione).

#### 3.2.2. SIRT1-Nrf2-ARE Pathway

Silent information regulator 2 homolog 1 (SIRT1) is a well-studied member of the Sirtuins family. It is widely expressed in the CNS, is also involved in the maintenance of physiological brain functions, and exhibits neuroprotective and anti-inflammatory effects in many neurodegenerative diseases [85,86]. Bidirectional crosstalk between SIRT1 and Nrf2 has been reported in human renal proximal tubular and glomerular mesangial cells [87,88]. Nuclear localization of Nrf2 is increased by SIRT1 through suppression of p53 in human mesenchymal stem cells [89]. It has been reported that melatonin can restore SIRT1 activity in the rat brain and protects brain from LPS-induced brain injury by activating SIRT1/Nrf2 signaling pathways [90]. More and more evidence has shown that SIRT1 activation can improve brain injury after aSAH and plays a key role in regulating the Nrf2 signaling pathway [91,92,93,94]. Salvianolic acid B is a nicotinamide adenine dinucleotide-dependent deacetylase and an effective SIRT1 activator [57,58], which is related to a variety of cell functions. In the past, the therapeutic mechanism of salvianolic acid B in aSAH has been unclear. Recently, the researchers found that the therapeutic effect of salvianolic acid B on EBI caused by aSAH largely depended on the upregulation of the Nrf2 signaling pathway, which was partly achieved by enhancing SIRT1 activation, rather than directly by regulating Keap1 [95,96].

Similarly, isoliquiritigenin (ISL) can improve EBI caused by aSAH through the Keap1-independent pathway. It has been proved that Isoliquiritigenin (ISL) has a strong antioxidant effect and prevents oxidative stress-related diseases [97,98]. In central nervous system (CNS) diseases, ISL has also been found to play a brain-protective role and it improves cognitive impairment in various acute brain injuries and neurodegenerative diseases [99,100,101]. In addition, ISL can penetrate the BBB and directly influence the central nervous system [100]. In this study, the mechanism of isoliquiritigenin in EBI caused by aSAH was further studied, and it was proved that ISL mediated brain protection through SIRT1 and increased the expression of Nrf2. At the same time, because of its high permeability through the BBB, it has a more potent effect on the central nervous system, which provides it with better clinical application potential. In addition, in the drug treatment of acute liver failure, studies have proved that isoliquiritigenin can improve the ability of antioxidant stress and reduce both inflammatory reaction and apoptosis by activating PGC1-α/Nrf2 signaling pathway, which provides the possibility of treating acute liver failure [102]. Therefore, isoliquiritigenin and puerarin have the ability to regulate PGC1-α/Nrf2 signaling pathway, while isoliquiritigenin has the potential to regulate Nrf2 through various Keap1-independent signaling pathways. Whether isoliquiritigenin can play a regulatory role through the PGC1-α/Nrf2 signaling pathway in aSAH treatment has yet to be definitively proven and requires additional research. 

In summation, the Keap1-independent Nrf2 regulatory compounds and Keap1-dependent Nrf2 regulatory compounds have similar effects on EBI after aSAH; both mainly upregulate the expression of antioxidant enzymes and detoxification enzymes. It is important to emphasize that some compounds, such as melatonin, can regulate Nrf2 expression in both Keap1-dependent and Keap1-independent ways. As previously mentioned, melatonin can improve EBI in aSAH through the Keap1-Nrf2-ARE pathway. In addition, studies have reported that melatonin can also reduce LPS-induced acute depressive-like behavior and microglial NLRP3 inflammasome activation through the SIRT1-Nrf2 pathway. Therefore, whether melatonin can also improve EBI after aSAH through the SIRT1-Nrf2 pathway and the specific mechanism of its regulation of Nrf2 are in need of further study.

## 4. Conclusions and Future Directions

In summary, when Nrf2 is used as a therapeutic target for aSAH: (1) it enhances the expression of antioxidant enzymes and detoxification enzymes mainly through the Nrf2-ARE pathway; (2) it can also enhance the metabolic efficiency of mitochondria and reduce the production of oxidative stress; and (3) it not only reduces the EBIs, but also improves cognitive function in the long term.

EBIs following aSAH leads to early neurological impairments, and most patients with aSAH also experience long-term cognitive impairment, which makes for a poor prognosis in the late stage. Nrf2 pathway targeted drugs can not only alleviate EBIs after aSAH, but also have the potential to treat chronic progressive cognitive dysfunction in patients with aSAH. Additional clinical studies are still needed to determine the short- and long-term efficacy of Nrf2 activators in patients with aSAH.

Nrf2 is a master regulator of a complex biological network of molecules and enzymes implicated in the regulation of antioxidant, drug metabolism, anti-inflammation, detoxification, and free radical scavenging. Targeting the Nrf2 pathway and reducing ROS levels is one potential neuroprotective strategy that may be implemented after aSAH. Since mitochondria are the main source of oxidative stress after aSAH, it is important to further study the regulatory mechanism of Nrf2 on mitochondrial function. Extensive research on various Nrf2 inducers and their different mechanisms of action could be fundamental to the design of more effective antioxidants. Complex drug, pharmacological and surgical combinations, or therapeutic interventions involving two or more different systems may be viable future therapies.

## Figures and Tables

**Figure 1 molecules-28-01747-f001:**
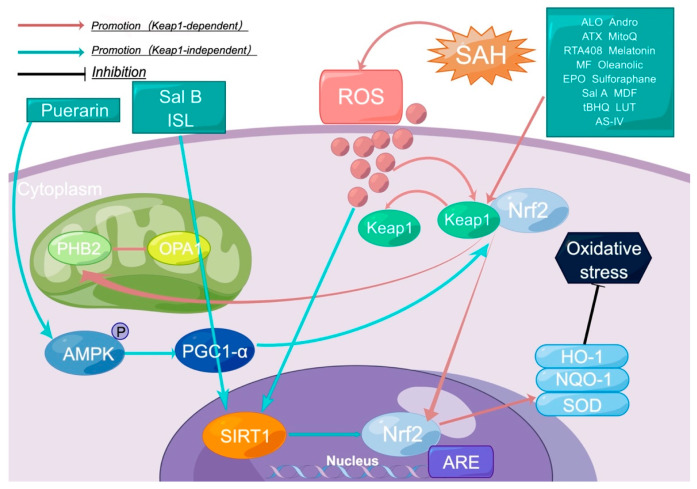
Mechanisms of Keap1-dependent and Keap1-independent pharmacological modulation of Nrf2 in aSAH. Created by Figdraw. (Sal B: salvianolic acid B; ISL: isoliquiritigenin; ALO: aloperine; Andro: adrographolide; ATX: astaxanthin; MF: mangiferin; EPO: erythropoietin; SAL A: salvianolic acid A; MDF: dimethyl fumarate; tBHQ: tert-butyl hydroquinone; LUT: luteolin; AS-IV: astragaloside; SAH: subarachnoid hemorrhage; ROS: reactive oxygen species; Nrf2: nuclear factor erythroid 2-related factor 2; Keap 1: kelchlike ech-associated protein 1; PHB2: prohibitin2; OPA1: optic atrophy 1; SIRT1: sirtuin 1; HO-1: heme oxygenase-1; NQO-1: quinine oxidoreductase-1; SOD: superoxide dismutase).

## Data Availability

Data sharing is not applicable to this article.

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
