# Peer review of "Pharmacological Modulations of Nrf2 and Therapeutic Implications in Aneurysmal Subarachnoid Hemorrhage"

_molecules, 2023, doi:10.3390/molecules28041747_

Round 1
Reviewer 1 Report
This manuscript describes the function of Nrf2 related to subarachnoid hemorrhage. More specifically, the mechanisms and the activator of Nrf2 were detailed. Overall, this manuscript is concise and thorough.
Concerns are:
Figure 1 should be mentioned in text, and the details should be included.
Some references are missing: for example, “In addition, studies have also shown that Nrf2-related 125 drugs can improve cognitive impairment after SAH and have certain value for the long- 126 term prognosis of patients with SAH.” needs references. And “studies have shown that some Nrf2 inducer 152 acting on Keap1-dependent pathway can improve cerebral vasospasm after SAH.” needs references, too.
Some typos – e.g., sulfhydryls should be sulfhydryl; certain value to a certain value, etc.
Author Response
Response to reviewers:
Reviewer #1
This manuscript describes the function of Nrf2 related to subarachnoid hemorrhage. More specifically, the mechanisms and the activator of Nrf2 were detailed. Overall, this manuscript is concise and thorough.
Response: Thank you very much for your high evaluation and positive comments. We have answered each of your points below.
Concerns are:
Figure 1 should be mentioned in text, and the details should be included.
Response: Many thanks for your comment. The description of the signaling pathways mentioned in Figure 1 has been added.
Nrf2-related aSAH drugs exert their neuroprotective effects mainly by up-regulating Nrf2 expression, through the Keap1-dependent or Keap1-independent pathway (Figure 1).
Some references are missing: for example, “In addition, studies have also shown that Nrf2-related 125 drugs can improve cognitive impairment after SAH and have certain value for the long- 126 term prognosis of patients with SAH.” needs references. And “studies have shown that some Nrf2 inducer 152 acting on Keap1-dependent pathway can improve cerebral vasospasm after SAH.” needs references, too.
Some typos – e.g., sulfhydryls should be sulfhydryl; certain value to a certain value, etc.
Response: Many thanks for your comment. We have revised the references and typos mentioned in the comment. We also have used English editing service to improve grammar errors.

Reviewer 2 Report
The manuscript entitled "Pharmacological Modulations of Nrf2 and Therapeutic Implications in Subarachnoid Hemorrhage" by Zhang et al. deals with the role that pharmacological compounds may have on the Nrf2 pathway during a subarachnoid hemorrhage. The manuscript is well articulated and seems complete in its different parts. I would suggest to include in paragraph 2 some more information regarding the immunomodulatory and neuroprotective role of Nrf2 pathway. Also, authors shall create a figure that encapsulates within it all the compounds they describe, this may help to improve the quality of the manuscript. English grammar and spell check is required as some periods may be too paratactic and difficult to understand.
Author Response
Reviewer #2
The manuscript entitled "Pharmacological Modulations of Nrf2 and Therapeutic Implications in Subarachnoid Hemorrhage" by Zhang et al. deals with the role that pharmacological compounds may have on the Nrf2 pathway during a subarachnoid hemorrhage. The manuscript is well articulated and seems complete in its different parts.
Response: Thank you very much for your high evaluation and positive comments. We have answered each of your points below.
I would suggest to include in paragraph 2 some more information regarding the immunomodulatory and neuroprotective role of Nrf2 pathway.
Response: Many thanks for your comment. We have added information about the immunomodulatory and neuroprotective role of the Nrf2 pathway as follows:
Rojo et al. found that nordihydroguaiaretic acid (NDGA) activates Nrf2 via MAPK and PI3K pathways and identified GSK-3β as an integrator of these pathways and a gatekeeper of Nrf2 stability at the Neh6 phosphorylation level21. Huang et al. found that PKC-catalyzed phosphorylation of Nrf2 at Ser-40 is a critical signaling event leading to ARE-mediated cellular antioxidant response 22.
Also, authors shall create a figure that encapsulates within it all the compounds they describe, this may help to improve the quality of the manuscript.
Response: Many thanks for your comment. Figure 1 included all the compounds mentioned in the context, and all the compounds have been placed according to their specific signaling pathway.
English grammar and spell check are required as some periods may be too paratactic and difficult to understand.
Response: Many thanks for your comment. We have corrected the typos and are using English editing service to improve grammar errors.

Reviewer 3 Report
The manuscript, which title is Pharmacological Modulations of Nrf2 and Therapeutic Implications in Subarachnoid Hemorrhage, is interesting. However, there are several questions in the manuscript.
1. The authors should provide the classification and epidemiology of Subarachnoid Hemorrhage.
2. The authors should provide the role of the Nrf2 pathway in the differential classification of Subarachnoid Hemorrhage.
3. The authors should provide the compounds of Nrf2 modulation in the differential classification of Subarachnoid Hemorrhage.
4. The authors should provide compounds of Nrf2 modulation in clinical patients or animals.
5. There are a lot of typo and grammar errors in the manuscript.
Author Response
Response to reviewers:
Reviewer #3
The manuscript, which title is Pharmacological Modulations of Nrf2 and Therapeutic Implications in Subarachnoid Hemorrhage, is interesting. However, there are several questions in the manuscript.
Response: Thank you very much for your high evaluation and positive comments. We have answered each of your points below.
- The authors should provide the classification and epidemiology of Subarachnoid Hemorrhage.
- The authors should provide the role of the Nrf2 pathway in the differential classification of Subarachnoid Hemorrhage.
- The authors should provide the compounds of Nrf2 modulation in the differential classification of Subarachnoid Hemorrhage.
Response: Thank you for the innovative advice from the reviewer. Considering the similarity of comments 1-3, we will answer your questions together here.
Generally, subarachnoid hemorrhage was classified by traumatic and spontaneous SAH, and 50%-80% of SAH happened due to the rupture of an intracranial aneurysm. Therefore, from the perspective of clinical treatment, targeted treatment for the aneurysm subarachnoid hemorrhage is more meaningful, and we revised the title and content of the review and focus on the Nrf2 modulation in the aneurysm subarachnoid hemorrhage (aSAH).
- The authors should provide compounds of Nrf2 modulation in clinical patients or animals.
Response: Many thanks for your comment. At present, there is still a lack of drugs for Nrf2 to treat subarachnoid hemorrhage in clinics, and studies based on experimental animals require further research about the practical application of related drugs.
- There are a lot of typo and grammar errors in the manuscript.
Response: Many thanks for your comment. We have revised the references and typos mentioned in the comment. We have corrected the typos and are using English editing service to improve grammar errors.

Reviewer 4 Report
Dear authors,
This review article depicts the mechanism of upregulation of Nrf2 via activators in Keap1-dependent or -independent pathways after Subarachnoid Hemorrhage (SAH). The abstract clearly points out the oxidative stress as a crucial cause of EBI and delayed CVS observed in SAH. References of Nrf2 upregulation against oxidative stress as well as for neuroprotection were well reviewed and summarized by category. The overview of the Nrf2 pathways in figure 1 was nicely generated de novo for an overall summary.
The overall context indeed delivered the message of the title while the brief of the mechanism in Keap1-independent pathway was lacking. The context of the Keap1-independent pathway would require more focus on mechanisms of the pathways, instead of descriptive results. The figure scheme did not thoroughly deliver a clear message of the mechanism regarding various factors “orchestrating” or regulating Nrf2 pathways.
Please carefully consider the following comments for your revisions:
1. Please revise section 2.2 for the mechanism of Keap1-independent pathway to concentrate on pathways involved in the activators of cited references for the regulation of expression level and function of Nrf2.
2. Please cite the reference(s) for the sentences (line 125-127, line 265-267).
3. Unfinished sentences (line 137, line 173-175).
4. Confusing sentences. Please rephrase it (line 145-150, line 291-293).
5. Sentence on line 183 with reference # 53 is inappropriate. This reference did not apply MitoQ in the study. Please consider removing this reference.
6. Please consider removing puerarin from line 189 of section 3.1. This activator was defined as in the Keap1-independent pathway by authors.
7. Please consider revising Table 1 and 2 as the following for clarity: i) Rearrange of the order of the compounds corresponding to the flow of the main context in section 3.1 and 3.2; ii) Capitalize the first letter of the description in the columns of Description and Therapeutic effects for easy reading; iii) Rename the column "Description" since the word "description" is vague.
8. Content of Table 1 for revision: i) Abbreviation for Dimethyl fumarate; ii) Inappropriate reference included in this category. This reference showed that the treatment of tBHQ induced autophagy in Nrf2 deficient mice. The autophagy triggered by tBHQ has no Keap1-Nrf2 involved. Please consider removing this reference.
9. Please consider giving a brief description of the Keap1-independent pathway in section 3.2.1 and 3.2.2 as you did with details for Keap1-dependent pathway in section 3.1.
10. For figure 1, please briefly describe each pathway and revise the figure with more specified signs of regulations as activation or inhibition. Please add figure 1 in the context.
11. Please include Table 1 and 2 in the context.
12. Please correct typos (line 68, line 140, line 259, line 264).
13. Please consider revising the conclusion to sum up all the information after reviewing cited references from the perspective of therapeutic implication in SAH.

Author Response
Thanks for your comments. Please find our responses in the attached file.

Round 2
Reviewer 1 Report
This manuscript has been much improved.
Reviewer 2 Report
The authors took into account my suggestions. The quality of the article significally improved. It can be accepted for publication in the present form.
Reviewer 3 Report
The manuscript is good enough to publication.